# Resveratrol May Reduce the Degree of Periodontitis by Regulating ERK Pathway in Gingival-Derived MSCs

**DOI:** 10.3390/ijms241411294

**Published:** 2023-07-10

**Authors:** Han Jiang, Jia Ni, Longshuang Hu, Zichao Xiang, Jincheng Zeng, Jiejun Shi, Qianming Chen, Wen Li

**Affiliations:** 1Stomatology Hospital, School of Stomatology, Zhejiang University School of Medicine, Zhejiang Provincial Clinical Research Center for Oral Diseases, Hangzhou 310000, China; 2Key Laboratory of Oral Biomedical Research of Zhejiang Province, Cancer Center of Zhejiang University, Hangzhou 310000, China; 3Engineering Research Center of Oral Biomaterials and Devices of Zhejiang Province, Hangzhou 310000, China; 4Stomatological Hospital, School of Stomatology, Southern Medical University, Guangzhou 510280, China; 5Dongguan Key Laboratory of Medical Bioactive Molecular Developmental and Translational Research, Guangdong Provincial Key Laboratory of Medical Molecular Diagnostics, Guangdong Medical University, Dongguan 523808, China

**Keywords:** gingival-derived MSCs (GMSCs), resveratrol, immunomodulation, periodontitis

## Abstract

Gingival-derived mesenchymal stem cells (GMSCs) have strong self-renewal, multilineage differentiation, and immunomodulatory properties and are expected to be applied in anti-inflammatory and tissue regeneration. However, achieving the goal of using endogenous stem cells to treat diseases and even regenerate tissues remains a challenge. Resveratrol is a natural compound with multiple biological activities that can regulate stem cell immunomodulation when acting on them. This study found that resveratrol can reduce inflammation in human gingival tissue and upregulate the stemness of GMSCs in human gingiva. In cell experiments, it was found that resveratrol can reduce the expression of TLR4, TNFα, and NFκB and activate ERK/Wnt crosstalk, thereby alleviating inflammation, promoting the proliferation and osteogenic differentiation ability of GMSCs, and enhancing their immunomodulation. These results provide a new theoretical basis for the application of resveratrol to activate endogenous stem cells in the treatment of diseases in the future.

## 1. Introduction

Periodontal tissues include gingiva, periodontal ligaments (PDLs), alveolar bone, and cementum. When there is acute (sometimes even aggressive) or chronic inflammation in periodontal tissue, we call it periodontitis. Periodontitis is related to adhesive dental plaque on the tooth surface, and is characterized by the progressive destruction and absorption of tooth supporting tissues (including PDLs and alveolar bone) [1,2], accompanied by gingival inflammation. The goal of periodontal therapy is to slow down or even arrest the progress of periodontitis, while regenerating the damaged periodontal tissue if possible [3]. At present, bone replacement materials are mostly used in regenerative periodontal surgery, but the indications are relatively limited. By contrast, the use of autologous mesenchymal stem cells (MSCs) for periodontal tissue regeneration is undoubtedly a more promising therapy method.

MSCs are actually stromal cells that have the capacity to self-renew and exhibit multilineage differentiation, which can proliferate in vitro as plastic-adherent cells, have fibroblast-like morphology, and form colonies in vitro [4,5]. MSCs can be isolated from a variety of tissues. Among them, a type of stem cell was isolated from the gingiva and named gingival-derived MSCs (GMSCs) [6]. Due to the easy access of gingival tissue from the oral cavity, GMSCs are more conveniently obtained compared to other stem cells. Meanwhile, GMSCs also have strong self-renewal, multilineage differentiation, and immunomodulatory/anti-inflammatory properties, which may play a meaningful role in the treatment of periodontitis [7]. Although most stem cell therapies using exogenous stem cells can bring certain therapeutic effects, stem cells are prone to aging during in vitro expansion and passage, and the early apoptosis rate of transplanted cells is high [8,9]. Therefore, regulating endogenous stem cells to repair damaged tissues has become a new research topic.

Resveratrol was initially considered to be only a plant antitoxin, which mainly exists in some vegetables, fresh fruits (such as mulberry, blueberry, and grape), and some Chinese medicinal materials (such as knotweed and Cassia tora), and is one of the polyphenol compounds produced by plants in response to environmental stress [10,11]. Subsequently, it was found that when applied to animals, it can show a variety of biological activities, including antioxidant [12], anti-inflammatory [13], cardiovascular protection [14], and anti-aging [15]. Resveratrol can reduce inflammation by inhibiting the activation of T cells [16] and NFκB signal transduction [17], and can also promote the proliferation and osteogenic differentiation of human bone marrow-derived MSCs (HBMSCs) by activating ERK1/2 pathway [18]. The application of resveratrol in animal models or clinical trials, to a certain extent, downregulates the expression of inflammatory factors in periodontal inflammatory tissue, and reduces the degree of periodontal damage [19,20,21].

Our previous research has shown that resveratrol can partially rescue bone loss in periodontitis by enhancing the immunomodulation of endogenous stem cells in mice [22]. However, it has not been verified in humans. Therefore, this study aimed to identify whether resveratrol can activate GMSC to play an anti-inflammatory role and promote tissue regeneration. In this study, we found that resveratrol could promote the proliferation, and osteogenic differentiation of GMSC, and play an immunomodulatory role. It was found that after being cultured with resveratrol, endogenous stem cells were significantly activated and the infiltration of inflammatory cells in human inflammatory gingival tissues was inhibited.

In brief, our data prove that resveratrol has a positive effect in the treatment of periodontitis, and provide a new theoretical basis for the application of resveratrol in the therapy of periodontal diseases in the future.

## 2. Results

### 2.1. Resveratrol Reduces Inflammation and Enhances Cell Stemness in Gingival Tissue

When resveratrol acts on periodontal tissue, it can downregulate the expression of inflammatory factors in the tissue [19]. Therefore, in order to further investigate the effect of resveratrol on inflammatory gingival tissue, we collected inflammatory gingival tissues from patients undergoing periodontal surgery, put them into a culture medium with or without resveratrol for 12 h, fixed the tissue, made paraffin sections, and stained them with HE. The results showed that the infiltration density of inflammatory cells in the resveratrol treatment group was significantly lower than that in the control group (Figure 1A).

Another part of the cultured gingiva was taken, and the total RNA was extracted for quantitative polymerase chain reaction assay (qRT-PCR) to further verify the effect of resveratrol on gingival tissue inflammation and to explore the changes in the expression of stemness genes. The results showed that after resveratrol treatment, the relative expression of the NFkB gene significantly decreased, while the relative expression of CD105 and CD90 increased.

We had known that GMSCs exist in the connective tissue beneath the gingival epithelium [23], so we used immunofluorescence staining to investigate the effect of resveratrol on human GMSCs (hGMSCs) in the gingiva. It could be seen that the fluorescence intensity of CD105 and CD90 (two surface molecular markers of MSC) in the resveratrol-treated group was significantly higher than that in the control group (Figure 1C,D). The above results demonstrated that resveratrol could reduce inflammation in the gingiva and enhance the stemness of hGMSCs.

### 2.2. Resveratrol Promotes hGMSCs Proliferation and Osteogenic Differentiation Capacity

Next, we proved the positive effect of resveratrol on the proliferation of hGMSCs. We added different concentrations of resveratrol to the culture medium and tested the cell proliferation with CCK-8 at 12 h, 24 h, and 36 h after adding the drug in order to explore the optimal concentration of resveratrol. The results showed that resveratrol at concentrations of 5 μM, 10 μM, 20 μM, and 50 μM could promote the proliferation of hGMSCs. However, compared with other concentrations, 10 μM resveratrol achieved the most significant effect (Figure 1E).

Then we put hGMSCs into the experimental group and control group, and cultured cells under osteogenic inductive medium with 10 μM resveratrol or an equal amount of dimethyl sulfoxide (DMSO). After 2 weeks, the total proteins were extracted for Western blot to compare the content of osteogenic differentiation markers between the two groups. We found that resveratrol significantly upregulated the expression of Runx2 and ALP (Figure 1F).

Similarly, after 3–4 weeks of culture, the cells were stained with alizarin red. The results showed that the resveratrol treatment groups had a larger number of red osteogenic calcification nodules than the DMSO treatment groups under the light microscope and by naked eye (Figure 1G).

These results indicated that resveratrol can promote the osteogenic differentiation of hGMSCs to a certain extent.

### 2.3. Resveratrol Enhances Stemness of hGMSCs In Vitro

Next, we used RNA sequencing (RNA-seq) technology to analyze the transcriptomic profiles of hGMSCs with or without resveratrol treatment. We found a total of 247 genes, which significantly changed their expressions,|log2 fold change (FC)|>1 and *p* < 0.05 (Figure 2A,B). Among these, 170 genes (68.8%) were upregulated, and 77 genes (31.2%) were downregulated in the experimental groups compared with the non-drug groups (Figure 2C). KEGG enrichment analysis (Figure 2D) showed that with resveratrol treatment, the enrichment of differentially expressed genes related to NFκB signaling pathway was significant, indicating that resveratrol may regulate hGMSC activity through NFκB signaling pathway.

Meanwhile, we also found that 35 stemness genes significantly changed their expressions after resveratrol treatment, including MSC markers (VCAM1, ICAM1), osteoprogenitor markers (SPP1, GLI1), chondroprogenitor markers (SOX9, ALCAM), and so on. Most of these genes were upregulated (Figure 1H). This showed that resveratrol might contribute to tissue regeneration and repair by promoting the stem cell characteristics of hGMSCs.

### 2.4. Resveratrol Activates ERK/Wnt Signaling Pathway in hGMSCs and Enhances FASL-Mediated Immunomodulation

The above results have already shown that resveratrol can promote osteogenic differentiation and the expression of stemness genes in hGMSCs. Next, we wanted to explore the specific pathways that resveratrol affected in hGMSCs. Since previous studies have shown that resveratrol can activate the ERK/Wnt signaling pathway of PDLSCs and thus affect their osteogenic differentiation ability [22], we speculated that there may be a similar pathway in hGMSCs. As expected, we found that the expression level of p-ERK increased after resveratrol treatment, indicating the activation of ERK signaling. At the same time, the expression levels of β-catenin and active β-catenin, which are related to canonical Wnt signaling, also increased (Figure 3A).

To further explore whether Wnt/β-catenin signaling pathway is a downstream pathway of ERK signaling, we transfected small interfering RNA (siRNA) into hGMSCs to silence the ERK gene and found that siERK treatment simultaneously reduced the level of p-ERK and activity β-catenin in hGMSCs, indicating that the Wnt/β-catenin signaling pathway is controlled by ERK signaling (Figure 3A). 

Fas ligand (FasL) is a type II transmembrane protein and is the common executioner of apoptosis within the tumor necrosis factor (TNF) family [24,25]. MSCs can induce activated T cell apoptosis via cell-cell contact by producing FAS ligand (FASL) [26]. To investigate whether resveratrol induces immunomodulation of hGMSCs by activating FASL through ERK signaling, we once again measured the level of FASL in hGMSCs with different treatments using a Western blot test. The results showed a significant increase in FASL level after resveratrol treatment, while siERK treatment decreased FASL level, indicating that resveratrol induced FASL-mediated immunotherapy of hGMSCs by activating the ERK signaling pathway (Figure 3A).

Collectively, our results revealed that resveratrol can activate the ERK/Wnt signaling pathway in hGMSCs and enhance FASL-mediated immunomodulation.

### 2.5. Resveratrol Inhibits the Inflammatory Pathway in hGMSCs

To further verify whether resveratrol can inhibit intracellular inflammatory pathways, we added lipopolysaccharide (LPS) to the culture medium to simulate periodontal inflammatory conditions, and then analyzed the activation of the inflammatory pathway by detecting the fluorescence intensities of TNFα and NFκB. It could be seen that the fluorescence intensities of TNFα and NFκB in the resveratrol-treated group were significantly lower than that of the control group (Figure 3C,D). It can be inferred that resveratrol may alleviate inflammatory response by downregulating the expression of TNFα and NFκB.

### 2.6. Resveratrol Is Associated with hGMSCs-Mediated Immunomodulation

Immunomodulatory properties are considered an important characteristic of stem cells, but there is currently no clear conclusion on whether resveratrol can affect hGMSCs-mediated immunomodulation. To investigate this issue, we co-cultured T cells with hGMSCs for 3 days and analyzed the apoptosis of T cells and the proportion of Th1 cells using flow cytometry. As shown in the figures, we found that regardless of whether the cells were treated with LPS or not, the proportion of FITC and PI double-positive cells significantly increased after the addition of resveratrol, indicating that resveratrol-treated hGMSCs can induce T cell apoptosis (Figure 4A).

Th1 cells are a subpopulation of T helper cells that secrete cytokines, such as IFNγ and TNFα, which can promote the activation of neutrophils and macrophages, thereby promoting the progression of local inflammation. Flow cytometry results showed that the proportion of CD4^+^IFNγ^+^ cells (Th1 cells) decreased with resveratrol treatment, which made it clear that resveratrol could to some extent reduce Th1 cell differentiation and reduce inflammation (Figure 4B).

### 2.7. TLR4 in hGMSCs Can Be Activated by LPS, and Resveratrol Can Reverse This Process

*Porphyromonas gingivalis* (*P. gingivalis*) is one of the major pathogens responsible for periodontitis progression. Lipopolysaccharide (LPS), a primary virulence factor of *P. gingivalis*, plays a strong pathogenic role in periodontal tissues [27,28]. Toll-like receptor 4 (TLR4) plays a critical role in the host inflammatory response facilitated by LPS. After LPS treatment, the immunofluorescence results showed a significant increase in TLR4 fluorescence intensity (Figure 4A,B). To investigate whether resveratrol treatment can rescue LPS-induced inflammatory response, we treated another group of cells with resveratrol for 24 h in advance. The results showed a significant decrease in TLR4 fluorescence intensity compared with the LPS group, indicating that LPS can activate TLR4 and promote inflammation, while resveratrol can reverse this process (Figure 4C,D).

## 3. Discussion

There have already been many treatment methods that utilize exogenous stem cells to promote bone regeneration and repair through direct injection or tissue engineering techniques [29,30]. However, compared to exogenous stem cells with a potential risk of immune rejection, autologous resident stem cells are undoubtedly the first choice for regenerative therapy. Resveratrol is a drug with great therapeutic potential that may improve the therapeutic effects of MSCs by enhancing their survival, proliferation, anti-inflammatory ability, and osteogenic differentiation. In this study, we investigated for the first time whether resveratrol can activate hGMSCs to exert anti-inflammatory effects and promote tissue regeneration. We found that resveratrol could promote the proliferation and osteogenic differentiation ability of hGMSCs, and active hGMSCs-mediated immunomodulation to reduce local inflammation. 

First, we collected human inflammatory gingival tissues, followed by culturing, sectioning, and staining. It was found that resveratrol treatment significantly reduced the infiltration of inflammatory cells in gingival tissue. Meanwhile, considering that hGMSC resides in the connective tissue layer of human gingiva, qRT-PCR and immunofluorescence staining were performed for MSC surface markers. We found that the fluorescence of MSC markers significantly increased after resveratrol treatment, indicating that the remission of gingival tissue inflammation may be related to the activation of hGMSC.

Then, we demonstrated that resveratrol can promote the proliferation of hGMSCs and revealed, through Western blot and alizarin red staining experiments, that resveratrol could promote osteogenic differentiation of hGMSCs at both the protein and macro levels, indicating that it may have a certain therapeutic effect on promoting bone regeneration in periodontitis.

Next, we used RNA sequencing (RNA-seq) technology to analyze the transcriptomic profiles of hGMSCs. RNA-Seq is a high-throughput sequencing method that can provide insight into the transcriptome of cells [31]. It can help us identify genes with significant changes in expression levels between cells of different treatment groups and, based on this, identify possible pathways of action. We found that 35 stemness genes significantly changed their expressions after resveratrol treatment, including (1) markers of angiogenesis (VCAM1, ICAM1), (2) markers of high osteogenicity (SPP1, GLI1), and (3) markers of high chondrogenicity (SOX9, ALCAM). These stemness genes with elevated expression suggested that resveratrol might exert the stem cell function of hGMSCs and repair tissue injury in these ways.

In the KEGG pathway enrichment analysis, we found that NFκB signaling pathway was enriched. As is well known, the NFkB signaling pathway is a classic inflammation-related pathway that interacts with multiple signaling molecules, indicating that the anti-inflammatory ability of resveratrol may be related to the regulation of NFκB signaling pathway.

Although resveratrol has been reported to inhibit the ERK signaling pathway when used in the treatment of cerebral hemorrhage or some tumors [32,33,34], studies have also shown that resveratrol can promote osteogenic differentiation of stem cells by activating the ERK pathway and Wnt/β-cantenin pathway [22,35]. The specific effect of inhibition or activation seems to depend on the concentration of resveratrol [36]. Meanwhile, the Wnt/β- catenin signaling pathway has also been confirmed to be positively regulated by ERK signaling. Therefore, we transfected siERK into the cells, demonstrating that resveratrol can regulate hGMSCs by activating the ERK/Wnt signaling pathway.

In addition, TNFα, NFκB, and TLR4 are all inflammatory-related pathways. TLR4 can be directly activated by LPS, triggering downstream inflammatory signaling pathways, such as TNFα signaling and NFκB signaling [37,38]. In cell experiments, we found that after resveratrol treatment, the expressions of these proteins were reduced, indicating that resveratrol can alleviate inflammation by affecting these pathways. However, further verification is needed to determine whether the downregulation of these inflammatory pathways can be regulated by the ERK signaling pathway.

The therapeutic effect of mesenchymal stem cells is mainly the result of immunomodulation mediated by inflammation. Generally speaking, MSCs secrete TGF-β, indoleamine 2,3-dioxygenase (IDO) (in humans) or inducible nitric oxide synthase (iNOS) (in mice), CXCR3 ligands, vascular cell adhesion molecule-1 (VCAM-1), and other immune regulatory factors, chemokines, and adhesion factors enable immune cells to gather near MSCs, thus effectively inhibiting the immune response [39]. For example, IDO can catalyze the transformation of tryptophan into L-caninic acid and pyridinecarboxylic acid, thereby inhibiting the proliferation of T cells [40]. Meanwhile, these tryptophan metabolites are more cytotoxic to Th1 than Th2, so they can promote the transformation of pro-inflammatory Th1 into anti-inflammatory Th2 [41]. 

Periodontitis is a multifactorial chronic inflammatory disease. We simulated the periodontal inflammatory environment by adding LPS to the culture medium in order to investigate the response of T lymphocytes to hGMSC-mediated immunomodulation. Th1 cells are a subpopulation of T helper cells that are predominantly distributed in periodontitis. They are responsible for producing various cytokines that maintain and amplify inflammation, and can promote the recruitment and permanence of active immune cells [42]. 

Our data showed that the proportion of Th1 cells was significantly reduced with resveratrol treatment, indicating that resveratrol can reduce inflammation. On the other hand, T cell apoptosis and the expression level of FASL in hGMSCs both increased, which made it clear that resveratrol promoted immunomodulation of hGMSCs and induced T cell apoptosis by activating FASL. Unfortunately, the comparison results between the control group and LPS-treated group were not ideal, which may be due to the concentration of LPS.

In summary, our data demonstrated that resveratrol can play a positive role in the treatment of periodontitis by promoting the proliferation, osteogenic differentiation, and immunomodulation of hGMSCs, thereby providing a new theoretical basis for the future application of resveratrol to activate endogenous stem cells in the treatment of periodontal diseases.

## 4. Materials and Methods

Cell culture. Human gingival-derived MSCs (hGMSCs) were obtained from Dongguan Key Laboratory of Medical Bioactive Molecular Developmental and Translational Research, Guangdong Medical University, Dongguan 523808, China. It was approved by the Medical Ethics Committee of Binhaiwan Central Hospital of Dongguan (No. 2021047). The hGMSCs were first planted for proliferation under alpha minimum essential medium (α-MEM, Basalmedia, Shanghai, China) supplemented with 10% fetal bovine serum (FBS, Biological Industries, Israel), 100 U/mL penicillin, and 100 U/mL streptomycin (Biosharp, Hefei, China) under a humidified atmosphere of 5% CO_2_ at 37 °C. hGMSCs in passages 3–5 were used in this study. 

Cell proliferation. The hGMSCs were seeded at 7000 cells/well in 96-well plates in complete culture medium for 12 h and were then treated with 5, 10, 20, and 50 μM resveratrol or dimethyl sulfoxide (DMSO) as vehicle control. At 12, 24, and 36 h after resveratrol treatment, cell proliferation was measured using the cell counting kit (CCK-8) (Yeasen Biotechnology, Shanghai, China) following the manufacturer’s specifications. Absorbance readings were then taken at 450 nm. Cell proliferation activity was calculated using the following formula: Cell proliferation activity (%) = [A (resveratrol) − A (blank)]/[A (control) − A (blank)] × 100.

In vitro osteogenic differentiation. For in vitro osteogenic differentiation assay, the hGMSCs (0.2 × 10^6^/well) were seeded in a 6-well culture plate and cultured under complete medium until the confluency of cells reached 80–90%. Then, the medium was changed to osteogenic inductive medium containing 0.05 mmol/L L-ascorbic acid (Macklin), 10 mM disodium β-glycerophosphate pentahydrate, and 10-7 M dexamethasone sodium phosphate (Solarbio, Beijing, China) with 10 μM resveratrol (Shifeng Biological, Shanghai, China) or equivalent volume of DMSO (dimethyl sulfoxide, Solarbio) treatment. After 2 weeks of induction, cultured cells were harvested for osteogenic marker analysis by Western blot. Cells were induced for 3–4 weeks, followed by alizarin red staining, to assess mineralized nodule formation.

ERK siRNA treatments. The ERK siRNA and control vehicle siRNA were obtained from Genepharma (Shanghai, China). The hDLSCs (1.8 × 10^5^/well) were seeded and cultured in a 6-well culture plate until 40% confluence, and then the culture medium was changed to reduced serum α-MEM for overnight, followed by siRNA transfection with Hieff Trans^®^ in vitro siRNA/miRNA transfection reagent (Yeasen Biotechnology) according to the manufacturer’s instructions. Cells were cultured with resveratrol treatment 24 h later and harvested for Western blot analysis after 48 h.

Western blot hybridization. Western blot analysis was performed as previously reported. Briefly, total protein was extracted using RIPA lysis buffer (Biosharp) with pmsf (phenyl methane sulfonyl fluoride, Solarbio) and phosphatase inhibitor cocktail (Abmole, Houston, TX, USA). Proteins were applied and separated on 4–12% precast protein plus gel (Yeasen Biotechnology), followed by transfer to nitrocellulose membranes (Biosharp). Membranes were then blocked by blocking buffer (Beyotime Biotechnology, Shanghai, China) for 1 h, followed by incubation with the primary antibodies at 4 °C for overnight. The primary antibodies used in this study included rabbit anti-FasL (1:500) (Bioworld, Dublin, OH, USA), anti- active β-Catenin (1:1000) (Cell Signaling Technology, Danvers, MA, USA), anti-β-catenin (1:8000) (Proteintech, Wuhan, China), anti- phospho-ERK1/2 (1:3000) (Proteintech), anti-ERK1/2 (1:1000) (Proteintech), anti- RUNX2 (1:1000) (Absin, Shanghai, China), and mouse anti-alkaline phosphatase (1:1000) (Genxspan, AL, USA). An HRP-conjugated secondary antibody (Proteintech; 1:20,000) was used to treat the membranes for 1 h. The protein bands were visualized using SuperSignal West Pico Chemiluminescent Substrate (Thermo Fisher Scientific, Waltham, MA, USA) and then documented using the ChemiDoc™ MP System (Bio-Rad, Hercules, CA, USA).

Collection and treatment of inflammatory gingival tissue. Inflammatory gingival tissues were collected from patients undergoing periodontal surgery. Gingival tissues were cut into two parts, and placed into culture medium supplemented with 10% FBS and 10 μM resveratrol or equivalent volume of DMSO for 12 h. This study was approved by the Ethics Committee of Stomatology Hospital, School of Stomatology, Zhejiang University School of Medicine (No. 2023–040). All patients gave written informed consent. 

RNA isolation and quantitative reverse transcriptase PCR (qRT-PCR) assays. Total RNA was isolated using the M5 Universal Plus RNA Mini Kit (Mei5bio, Beijing, China) from the cultured gingival samples according to the manufacturer’s instructions. The concentration of total RNA was measured using NanoDrop (thermo). The HiScript III 1st Strand cDNA Synthesis Kit (Vazyme, Nanjing, China) was used to prepare the cDNA. qRT-PCR assays were performed using ChamQ Universal SYBR qPCR Master Mix (Vazyme) and gene-specific primer pairs. The primers used in this study were synthesized and purified by Sangon Biotech (Shanghai, China). The RNA expression was normalized to GAPDH. QuantStudio™ 7 Flex qRT-PCR System (Thermo Fisher) was used for qRT-PCR analysis.

Haematoxylin and eosin (HE). Cultured gingival samples were fixed in 4% paraformaldehyde for 24 h, embedded in paraffin wax after dehydration, and sectioned at 7 µm. HE staining was used (Haokebio, Hangzhou, China) for histological analysis.

Immunofluorescent staining. Cultured gingival samples were fixed in 4% paraformaldehyde for 24 h, embedded in paraffin wax after dehydration, and sectioned at 7 µm. The sections were deparaffined with xylene and rehydrated with ETOH and ddH2O. After antigen retrieval, permeabilization, and block, the sections were incubated with primary antibodies including rabbit anti-CD105 (1:1000) (Proteintech), rabbit anti-CD90 (1:200) (Absin) at 4 °C overnight, followed by treating with secondary antibody (1:400) conjugated with Alexa fluor 488 for 1 h at room temperature. Finally, the cell nuclei were stained with DAPI (Solarbio). The fluorescence intensity was measured using Image J software.

Immunocytochemistry. The hGMSCs were seeded in a 12-well culture plate and cultured under complete medium until the confluency of the cells reached 50%. They were then divided into a control group, an LPS group, and an LPS + R group. In the LPS + R group, hGMSCs were treated with 10 μM resveratrol, and 24 h later, LPS (InvivoGen, Hongkong, China) with a final concentration of 100 ng/mL was added in the LPS and LPS + R groups. After another 24 h, hGMSCs were fixed with 4% paraformaldehyde, permeabilized with 0.5% Triton X-100, and blocked with 1% BSA. Primary polyclonal antibody rabbit anti-TLR4 (1:300) (Affinity Biosciences, Cincinnati, OH, USA, rabbit anti-TNFα (1:700) (GeneTex, Irvine, CA, USA), and rabbit anti-NFκB (1:100) (Bioworld) were used, followed by Alexa fluor 488 green fluorescence conjugated donkey anti-rabbit as secondary antibodies (1:500) (Absin). Finally, the cell nuclei were stained with DAPI (Solarbio). The fluorescence intensity was measured using Image J V1.52 software.

RNA isolation and RNA-seq. The hGMSCs in passage 3 were cultured with 10 μM resveratrol or equivalent volume of DMSO when the confluency of cells reached 60–70%. 24 h later, total RNA was isolated by using HK zol (Haokebio) reagent from the cultured cells according to the manufacturer’s instructions. The RNA samples were put into liquid nitrogen for rapid freezing and then placed at −80 °C, waiting for delivery to Shanghai OE Biotech Technology Co., Ltd. (Shanghai, China), for RNA-seq procedure and analysis. We also used the website provided by OE Biotech for figure drawing. The differential expression between conditions was statistically assessed, and genes with |log2 (FoldChange)| > 1 and *p* value < 0.05 were identified as differentially expressed. The enriched pathways of differentially encoded protein genes were analyzed by using the KEGG database and were sorted by the–log10 *p* value from largest to smallest to find possible related pathways.

Isolation and culture of mouse spleen lymphocytes. C57BL/6 mice were euthanized by cervical dislocation, and the spleens were taken. This study was approved by the Institutional Animal Care and Use Committee (IACUC), Zhejiang Center of Laboratory Animals (ZJCLA) (No. ZJCLA-IACUC-20050036). The operation was carried out according to the instructions of the mouse spleen lymphocyte extraction kit (TBD Science, Tianjin, China), and the obtained T lymphocytes were pre-stimulated with plate-bound anti-CD3e (5 μg/mL) and soluble anti-CD28 (2 μg/mL) antibodies for 3 days in RPMI Medium 1640 (Basalmedia) with 10% FBS (Biological Industries), 100 U/mL penicillin and 100 U/mL streptomycin (Biosharp). The hGMSCs were seeded in a 12-well culture plate and cultured under complete medium until the confluency of the cells reached 50%. Then they were divided into a control group, resveratrol group, LPS group, and LPS+R group, with or without resveratrol (10 μM) treatment for 6 h and followed by LPS (100 ng/mL) treatment for 24 h. Then, activated T lymphocytes (1 × 10^6^ per well) were directly loaded onto hGMSCs and co-cultured for 3 days.

Flow cytometry analysis. Th1 cells were stained with CD4 antibody conjugated with PerCP/Cyanine5.5 (Biolegend, San Diego, CA, USA) and IFNγ antibody conjugated with APC (biolegend), while apoptotic T cells were stained using Annexin V-FITC/PI Apoptosis Detection Kit (Yeasen Biotechnology), and then analyzed by flow cytometer (Beckman Coulter CytoFLEX) using CytExpert V2.5.0.77 software.

Statistical analysis. Each group in each experiment contained at least three samples, and each sample was set with at least three technical replicates. An unpaired two-tailed Student’s *t*-test was used to analyze statistical difference. *p* values less than 0.05 were considered statistically significant.

## Figures and Tables

**Figure 1 ijms-24-11294-f001:**
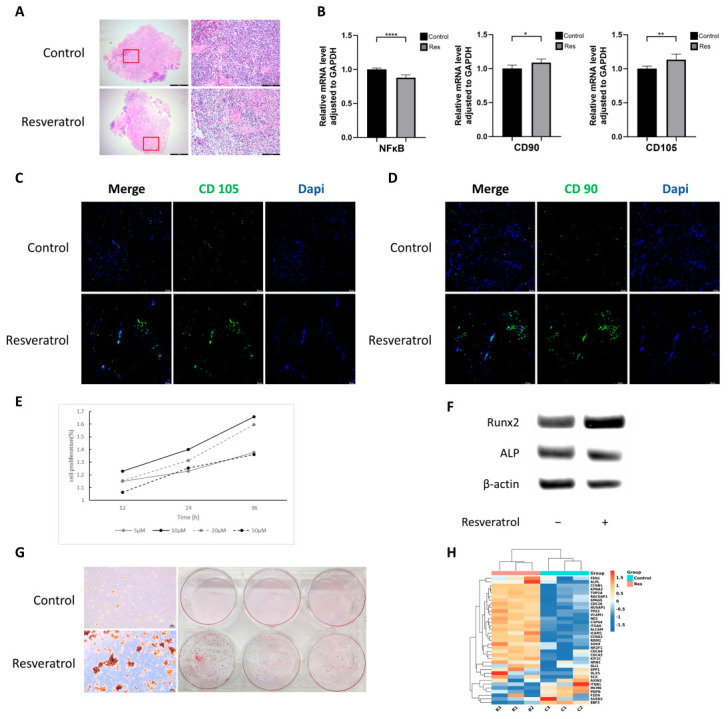
Resveratrol enhances the stemness of human gingival-derived MSCs (hGMSCs). (**A**) Histological examination of human inflammatory gingiva tissue cultured in medium with or without resveratrol for 12 h. Each region selected by a red rectangle in the left column pictures is enlarged in the right column(hematoxylin and eosin staining, magnification ×40 (left) and ×200 (right)). (**B**) Quantitative polymerase chain reaction assay showed the levels of inflammation-related and stemness genes with or without resveratrol treatment in human inflammatory gingiva tissue. The immunofluorescence of human inflammatory gingival tissue showed an increase in the fluorescence intensity of stem cell surface markers, such as CD105 (**C**) and CD90 (**D**), in gingival connective tissue treated with resveratrol (magnification × 200). (**E**) The cell proliferation curve was detected with CCK-8. It showed that when the final concentration was 10 μM, resveratrol could promote the proliferation of hGMSCs to the maximum extent. (**F**) Expression levels of the osteogenic markers Runx2 and ALP were verified through Western blot. (**G**) Alizarin red staining showed the capacity to form mineralized nodules under osteoinductive conditions with or without resveratrol treatment. Images were taken by an optical microscope under 40× magnification (left) and a camera (right). (**H**) Cluster heatmap showed hGMSCs stemness genes that were differentially regulated following resveratrol treatment. * *p* < 0.05, ** *p* < 0.01, **** *p* < 0.0001.

**Figure 2 ijms-24-11294-f002:**
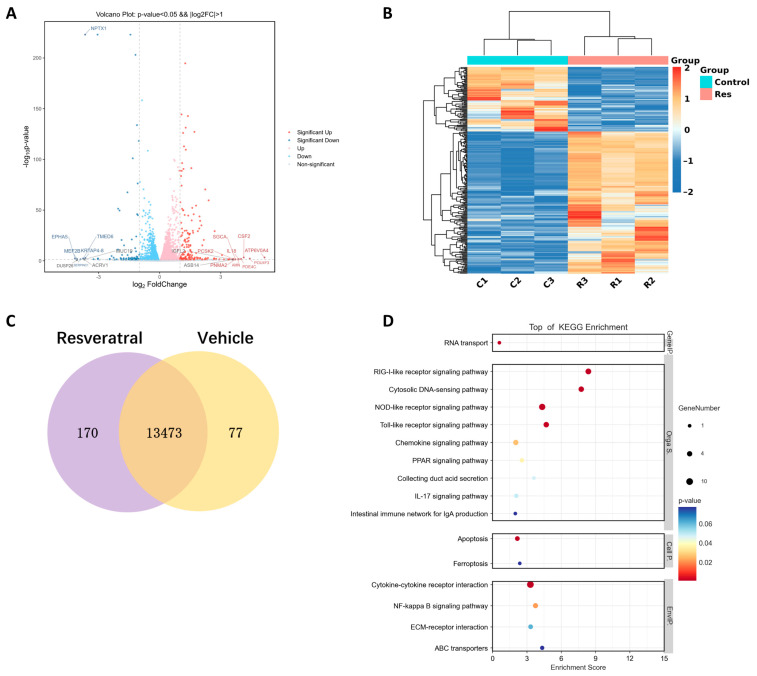
RNA-seq analysis showed resveratrol enhances the stemness of hGMSCs in vitro. (**A**) Volcano plot showed the distribution of differentially expressed genes between the two groups of samples, with the horizontal axis representing the log2-fold changes in gene expression after resveratrol treatment and the vertical axis representing statistical significance. (**B**) Cluster heatmap showed the genes that were differentially regulated following treatment with 10 μM resveratrol. (**C**) Venn diagram showed the number of differentially expressed genes in hGMSCs with 10 μM resveratrol treatment (*p* < 0.05). (**D**) Enrichment analysis of the KEGG pathway showed several signaling pathways with obvious enrichment.

**Figure 3 ijms-24-11294-f003:**
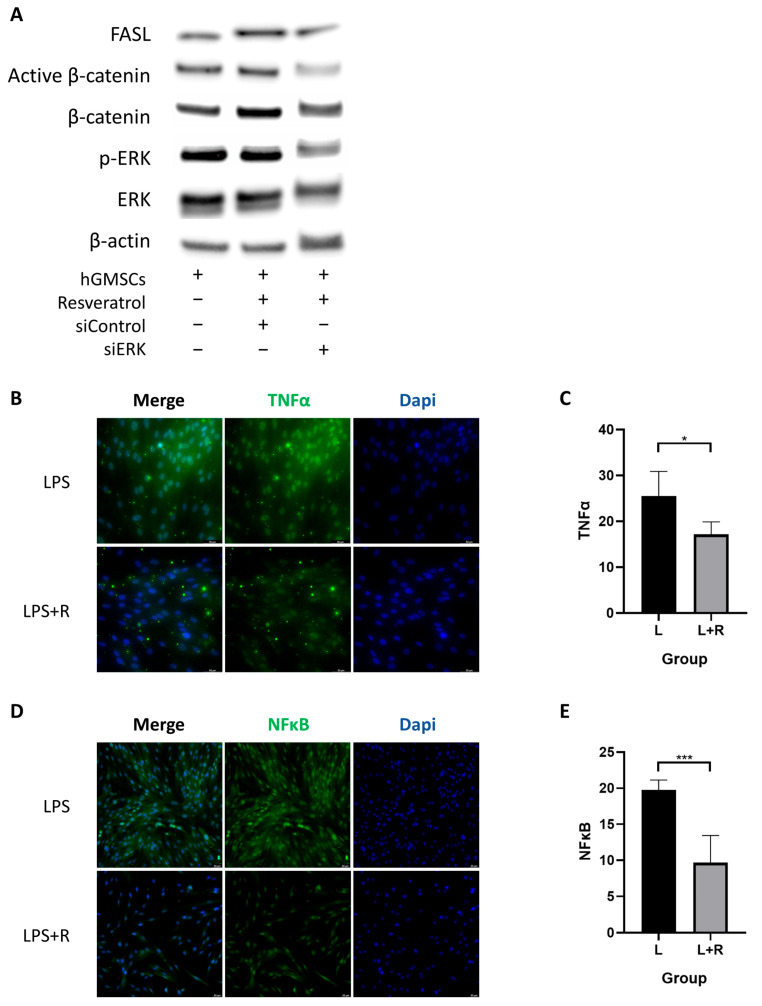
Resveratrol activated ERK/Wnt crosstalk and inhibited inflammatory pathways in hGMSCs. (**A**) Western blot analysis showed the expression levels of FASL, active β-catenin, β-catenin, p-ERK, and ERK in hGMSCs with different treatments. Immunofluorescence staining and the histogram of the fluorescence intensity revealed that the expression of TNFα (**B**,**C**) and NFκB (**D**,**E**) decreased in hGMSCs with resveratrol treatment (magnification ×200). * *p* < 0.05, *** *p* < 0.001.

**Figure 4 ijms-24-11294-f004:**
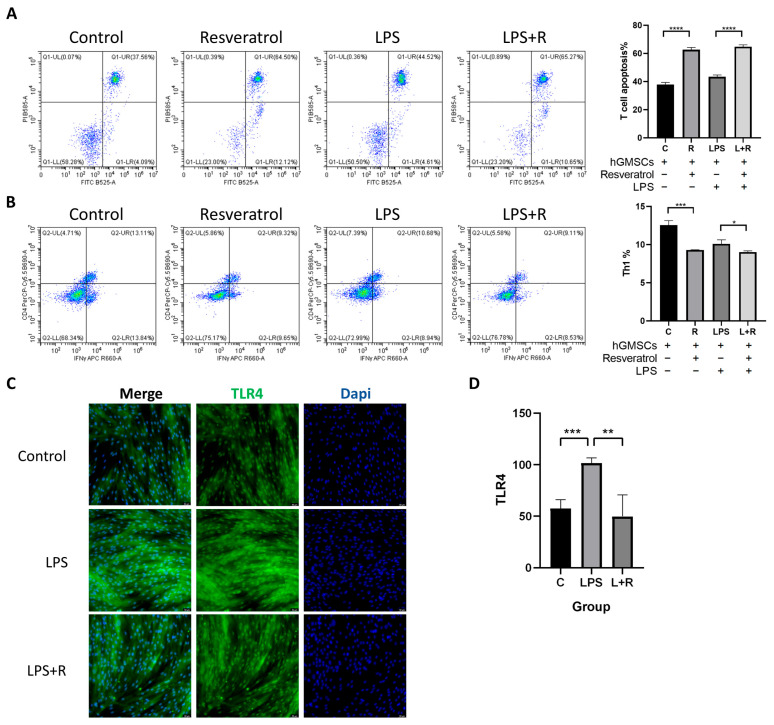
Resveratrol was associated with hGMSC-mediated immunomodulation. (**A**) Flow cytometry analysis showed that FITC^+^ PI^+^ cells (apoptotic T cells) increased with resveratrol treatment. (**B**) Flow cytometry analysis showed that CD4^+^ IFNγ^+^ cells (Th1 cells) decreased with resveratrol treatment. (**C**) Immunofluorescence staining showed the expression of TLR4 in hGMSCs with different treatments (magnification ×200). (**D**) Histogram of TLR4 fluorescence intensity. * *p* < 0.05, ** *p* < 0.01, *** *p* < 0.001, **** *p* < 0.0001.

## Data Availability

The data presented in this study can be available on request from Jiang H.

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
