# Peer review of "Resveratrol May Reduce the Degree of Periodontitis by Regulating ERK Pathway in Gingival-Derived MSCs"

_ijms, 2023, doi:10.3390/ijms241411294_

Round 1

Reviewer 1 Report

The study discusses the potential use of resveratrol to activate endogenous stem cells in the treatment of periodontal disease. They found that resveratrol can reduce inflammation in human gingival tissue and upregulate the stemness of GMSCs in human gingiva. The authors conclude by stating that these results provide a new theoretical basis for the application of resveratrol to activate endogenous stem cells in the treatment of diseases in the future. However, major changes are needed before publishing the manuscript. My comments are as follows:

1.     In the introduction, the authors should introduce periodontal tissues and gingival stem cells better. The term periodontitis is vaguely defined without introducing the periodontal tissues.

2.     Please provide the ethical information, if available, for the source of GMSCs.

3.     For the inflammatory gingival tissue collected from patients, please provide the IRB approved study/protocol number if available.

4.     Please provide additional details about the RNASeq analysis. The methods section mention the use of a website, but it is not clear how the RNASeq analysis was performed.

5.     The methods section mentions that an unpaired two-tailed Student's t-test was used for statistical analysis, but it does not provide information on the sample size or the number of replicates.

6.     In the results showing the Resveratrol treated inflammatory tissue (Fig 1A), there is stark difference in the size of the tissue imaged. The region of interest for quantification also seems to have been chosen in the area with fewer cells. Please provide a more comparable image from the treated group.

7.     The is quantification in the number of lymphocytes (Fig 1A), but it is not clear how many replicates were used. Was this data from one sample per group?

8.     In Fig 1B and 1C, there is lack of Dapi signaling in the control slides. The lack of CD105 and CD90 staining in control slides may be associated with lack of cells in the imaged area. Please provide better images for CD105 and CD90 stainings

9.     The association between FASL  (Figure 3) and resveratrol is not clear. The authors first mention that FASL has role in autophagy and later state it is involved in immunomodulation which are two very separate processes. More justification needs to be added for use of FASL.

Minor revisions and spell check is recommended

Author Response

Dear Reviewers:

Thank you for your letter and for the reviewers’ comments concerning our manuscript entitled “Resveratrol may Reduce the Degree of Periodontitis by Regulating ERK Pathway in Gingival-Derived MSCs”. Those comments are all valuable and very helpful for revising and improving our paper, as well as the important guiding significance to our researches. We have studied comments carefully and have made correction which we hope meet with approval. Revised portion are marked in red in the paper. The main corrections in the paper and the responds to the reviewer’s comments are as flowing:

Our paper have been professionally edited for English language by a native speaker.

  1. In the introduction, the authors should introduce periodontal tissues and gingival stem cells better.

Respond: Thank you for your suggestion. The introduction of periodontal tissues and gingival stem cells has been added in the revised manuscript. (Line 32-34, 47-56)

  1. The ethical information for the source of GMSCs had better to be mentioned if available.

Respond: The ethical information for the source of GMSCs has been added in the revised manuscript. (Line 346-348, 477-478)

  1. The IRB approved study/protocol number had better to be mentioned if available.

Respond: Because our study did not directly intervene in humans, but only collected inflammatory gingival removed from periodontal surgery, there is no need for IRB.

  1. Please provide additional details about the RNASeq analysis.

Respond: The additional details about the RNASeq analysis have been added in the revised manuscript. (Line 436-441)

  1. The methods section does not provide information on the sample size or the number of replicates.

Respond: The information of the sample size or the number of replicates have been added in the revised manuscript. (Line 460-461)

  1. In the results showing the Resveratrol treated inflammatory tissue (Fig 1A), there is stark difference in the size of the tissue imaged. The region of interest for quantification also seems to have been chosen in the area with fewer cells. Please provide a more comparable image from the treated group.

Respond: We have replaced the HE images in Fig 1A. (Line 137-152)

  1. The is quantification in the number of lymphocytes (Fig 1A), but it is not clear how many replicates were used. Was this data from one sample per group?

Respond: We took three different fields of view of one gingival sample per group at a magnification of 400x, counted lymphocytes, and averaged the number of lymphocytes. This method may not be very accurate, so it has been deleted in the revised manuscript.

  1. The lack of CD105 and CD90 staining in control slides may be associated with lack of cells in the imaged area. Please provide better images for CD105 and CD90 stainings.

Respond: In fact, in the immunofluorescence staining images of CD105 and CD90, Dapi signaling in the control group is more than that in the resveratrol treatment group, but the fluorescence signals of CD105 and CD90 are stronger in the resveratrol treatment group, so we think these images could confirm our conclusions. However, we agree with your viewpoint that we should try to unify the cell numbers of the two groups as much as possible. But due to the tight schedule, we did not replace them, and these pictures can also prove our conclusion.

  1. The association between FASL (Figure 3) and resveratrol is not clear. The authors first mention that FASL has role in autophagy and later state it is involved in immunomodulation which are two very separate processes. More justification needs to be added for use of FASL.

Respond: Very good suggestion. Our experiment does not involve autophagy, so we have removed this section. (Line 198-199)

Reviewer 2 Report

The objective of this study was to investigate the effects of Resveratrol on osteogenic differentiation and inflammatory response in GMSCs, as well as the underlying mechanisms involved. While the study identified some endpoints, the exploration of causal relationships was relatively limited. Therefore, further investigations are recommended to establish more comprehensive causal links. Below are some comments/suggestions for the authors to consider.

In general, all the figures are so blurred, and the author is advised to ensure the resolution of all the images and to enlarge the words in the figures. It is important to ensure that figures in a research paper are clear and easily interpretable. If the figures in the manuscript appear blurred or have low resolution, the author should take steps to enhance the image quality and make them more legible for readers.

In Fig 1, the authors should consider performing additional qPCR analysis of inflammatory cytokines in periodontitis to complement their evaluation of the anti-inflammatory effect of Resveratrol based on the examination of inflammatory cell infiltration in HE sections. While HE sections provide valuable histological information, they cannot be used as a quantitative analysis. Assessing the expression levels of inflammatory cytokines through qPCR would provide a more comprehensive understanding of the anti-inflammatory response induced by Resveratrol in periodontitis. This approach would allow for the quantification of specific inflammatory markers and provide a molecular-level assessment of the inflammatory modulation.

Similarly, immunofluorescence staining alone may not provide quantitative analysis. Therefore, the authors are advised to complement their findings with techniques such as western blotting or qPCR to quantify the results. While immunofluorescence staining allows for visualizing the presence and localization of proteins, it is not a quantitative method. By utilizing western blotting or qPCR, the authors can obtain quantitative data on the expression levels of these markers, proteins, or cytokines, providing a more accurate assessment of their presence and potential changes induced by experimental conditions.

In the Materials and Methods section, the authors mentioned using the CCK-8 reagent to measure the absorbance, possibly related to Formazan, for assessing cell proliferation. However, in Figure 1D, the results are presented as percentages rather than absorbance values. To resolve this discrepancy, the authors should provide clarification on how the absorbance data were converted into percentage values and explain the calculation method employed.

The Western blotting of β-actin observed in this study appears to be inhomogeneous, suggests a potential issue with the standardization of sample protein concentrations prior to performing Western blotting. To ensure accurate and reliable results, it is crucial to standardize protein concentrations among samples to account for variations in loading.

There are no scale bars are shown for Alizarin stains. Additionally, it is recommended that the authors measure osteocalcin levels and calcium deposition separately to gain a more detailed understanding of the effects of Resveratrol on osteogenesis. This would provide a more complete picture of the effects of Resveratrol on bone formation.

The plus sign for cells in Fig 3A and Fig 4 A is not necessary or potentially confusing, it is advisable for the author to consider removing it to enhance the clarity of the figure.

The authors attempted to illustrate the relationship between the Wnt/ERK signaling pathway and Resveratrol, but only assessed the expression of β-catenin. While this is a useful first step, it may not be sufficient to fully understand the relationship. To provide more comprehensive evidence, it is recommended that the authors use Western blotting to determine changes in protein levels, including Wnt, GSK-3β, and β-catenin. Additionally, it is recommended that the authors use inhibitors of the Wnt signaling pathway to determine the relationship between Resveratrol and this pathway. This would provide further evidence of the interaction between Resveratrol and the Wnt signaling pathway and help to confirm the authors' findings.

In addition, in order to investigate the relationship between the differentiation-enhancing and anti-inflammatory effects of Resveratrol and the Wnt/ERK pathway more comprehensively, it is advisable for the authors to re-evaluate the mentioned items by incorporating siRNA experiments instead of solely relying on Western blotting to determine protein changes.

The representation of the flow cytogram in Fig 4 as a histogram is recommended to improve visibility and facilitate the comparison between different conditions. Histograms provide a clearer visualization of the distribution and intensity of the analyzed parameter, allowing for easier identification of differences between groups. In contrast, dot plots may not effectively demonstrate any statistical gaps between the groups.

None.

Author Response

Dear Reviewers:

Thank you for your letter and for the reviewers’ comments concerning our manuscript entitled “Resveratrol may Reduce the Degree of Periodontitis by Regulating ERK Pathway in Gingival-Derived MSCs”. Those comments are all valuable and very helpful for revising and improving our paper, as well as the important guiding significance to our researches. We have studied comments carefully and have made correction which we hope meet with approval. Revised portion are marked in red in the paper. The main corrections in the paper and the responds to the reviewer’s comments are as flowing:

Our paper have been professionally edited for English language by a native speaker.

  1. In general, all the figures are so blurred, and the author is advised to ensure the resolution of all the images and to enlarge the words in the figures.

Respond: We have replaced all the figures, checked the resolution of the images, and enlarged some of the words.

  1. In Fig 1, the authors should consider performing additional qPCR analysis of inflammatory cytokines in periodontitis to complement their evaluation of the anti-inflammatory effect of Resveratrol based on the examination of inflammatory cell infiltration in HE sections.

Respond: We detected the expression level of NFκB inflammatory genes in gingival tissue through qRT-PCR and added the results to the revised manuscript. (Line 91-96, 137-152, 398-406)

  1. The authors are advised to complement their findings with techniques such as western blotting or qPCR to quantify the results.

Respond: Thank you for your suggestion. We also believe that some other experiments should be added to better validate the results. But we found it difficult to complete these experiments in such a short time, so we only detected the expression levels of CD105 and CD90 in gingival tissue through qRT-PCR and added the results to the revised manuscript. (Line 91-96, 137-152, 398-406)

  1. In the Materials and Methods section, the authors mentioned using the CCK-8 reagent to measure the absorbance for assessing cell proliferation. The authors should provide clarification on how the absorbance data were converted into percentage values and explain the calculation method employed.

Respond: The calculation method has been added in the revised manuscript. (Line 356-358)

  1. The Western blotting of β-actin observed in this study appears to be inhomogeneous, suggests a potential issue with the standardization of sample protein concentrations prior to performing Western blotting.

Respond: the β-actin observed in Western blotting may be inhomogeneous, which is generally difficult to determine the experimental results. However, in our experiment, due to the variation trends of β-actin and target proteins between the two groups are opposite, we can still draw the conclusions. (For example, the content of β-actin in group A is higher, but the content of the target protein in this group is lower, so it can still be proved that the content of the target protein in group A has decreased.)

  1. There are no scale bars are shown for Alizarin stains. Additionally, it is recommended that the authors measure osteocalcin levels and calcium deposition separately to gain a more detailed understanding of the effects of Resveratrol on osteogenesis.

Respond: Thank you for your suggestion. We have already added the scale bars of Alizarin stains in Figure 1G (Line 137). However, due to time constraints, we were unable to complete another osteogenic differentiation in time,so we replaced it by consulting the references. At present, many researches have proved that resveratrol can promote stem cell osteogenesis, so we did not conduct in-depth research[1-3].

[1] Dai, Z., et al., Resveratrol enhances proliferation and osteoblastic differentiation in human  mesenchymal stem cells via ER-dependent ERK1/2 activation. Phytomedicine, 2007. 14(12): p. 806-14.

[2] Wang, Y.J., et al., Resveratrol enhances the functionality and improves the regeneration of  mesenchymal stem cell aggregates. Exp Mol Med, 2018. 50(6): p. 1-15.

[3] Dosier, C.R., et al., Resveratrol effect on osteogenic differentiation of rat and human adipose derived stem cells in a 3-D culture environment. J Mech Behav Biomed Mater, 2012. 11: p. 112-22.

  1. The plus sign for cells in Fig 3A and Fig 4A is not necessary or potentially confusing, it is advisable for the author to consider removing it to enhance the clarity of the figure.

Respond: Our consideration is that these plus signs may allow readers to have a clearer understanding of the processing of each group when reading the figure. And considering that the figure may be vague, we have enlarged the font.

  1. The authors attempted to illustrate the relationship between the Wnt/ERK signaling pathway and Resveratrol, but only assessed the expression of β-catenin. While this is a useful first step, it may not be sufficient to fully understand the relationship. To provide more comprehensive evidence, it is recommended that the authors use Western blotting to determine changes in protein levels, including Wnt, GSK-3β, and β-catenin. Additionally, it is recommended that the authors use inhibitors of the Wnt signaling pathway to determine the relationship between Resveratrol and this pathway.

Respond: The apoptosis pathway we studied is ERK-Wnt FasL, and Wnt is located downstream of ERK. We have validated the relationship between ERK-Wnt signaling through siRNA, so there is no need to independently verify the Wnt signaling pathway. In addition, the time for revising is only 10 days, so there is almost no time to verify.

  1. It is advisable for the authors to re-evaluate the mentioned items by incorporating siRNA experiments instead of solely relying on Western blotting to determine protein changes.

Respond: The pathway we want to explore in this experiment is ERK/Wnt/FASL pathway. At present, we have preliminarily proved the influence of ERK signaling on downstream pathway through siERK.

  1. The representation of the flow cytogram in Fig 4 as a histogram is recommended to improve visibility and facilitate the comparison between different conditions.

Respond: Thank you for your suggestion. Histograms indeed provide a clearer visualization of the distribution and intensity of the analyzed parameter. But considering that this experiment used flow cytometry to filtrate double positive cells, compared to histograms, we think that dot plots could help readers more intuitively know the proportion of target cells, and there are also many articles using dot plots. However, the suggestion is also quite reasonable, so we changed dot plots into pseudo color plots to better display the number of target cells.

Round 2

Reviewer 1 Report

The authors have responded to my questions.

Author Response

Thank you very much for your affirmation and approval to the revised manuscript. And thank you again for taking the time and effort to review our manuscript.

Reviewer 2 Report

The authors found answers to many of their requests.

However, they are unclear about the immunostained images. Please solve this image problem.

Author Response

Thank you for the comment concerning our manuscript. We have rechecked the immunofluorescence images in the manuscript and speculated that you may think the images of CD105 and CD90 in Figure 1 are not clear. Therefore, we increased the fluorescence brightness in these images, hoping to ensure a clearer fluorescence display while ensuring the trend of fluorescence intensity of different groups are the same as before. (Line 137)